Serum cystatin-C and all-cause mortality in patients with hypertrophic cardiomyopathy: a retrospective cohort study

Liu Lu 1
Zheng Yi 1
Ruan Haiyan 2
Wang Ziqiong 1
Chen Xiaoping xiaopingchen0196@163.com 1
He Sen hesensubmit@163.com 1 3
1 Department of Cardiology, West China Hospital of Sichuan University , Chengdu , Sichuan , China
2 Department of Cardiology, Hospital of Traditional Chinese Medicine, Shuangliu District , Chengdu , Sichuan , China
3 Karamay Hospital of Integrated Chinese and Western Medicine , Karamay , Xinjiang Uygur Autonomous Region , China
Barbosa Neto Octavio
Electronic publication date: 2025 Nov 7
Publication date: 2025
Volume: 13
Electronic Location ID: e19631
Received 2023 Nov 7; Accepted 2025 May 30
Copyright: ©2025 Liu et al.
Copyright year: 2025
Copyright holder: Liu et al.
License: This is an open access article distributed under the terms of the Creative Commons Attribution License, which permits unrestricted use, distribution, reproduction and adaptation in any medium and for any purpose provided that it is properly attributed. For attribution, the original author(s), title, publication source (PeerJ) and either DOI or URL of the article must be cited.
License URL: https://creativecommons.org/licenses/by/4.0/

Keywords: Hypertrophic cardiomyopathy, Mortality, Cystatin-C, eGFR, Creatinine

Funding: Sichuan Science and Technology Program, China 2022YFS0186 the National Natural Science Foundation of China 81600299 The Sichuan Science and Technology Program 2023NSFSC1632 The study was funded by Sichuan Science and Technology Program, China (Grant No. 2022YFS0186), the National Natural Science Foundation of China (Grant No. 81600299), and the Sichuan Science and Technology Program (2023NSFSC1632). The funders had no role in study design, data collection and analysis, decision to publish, or preparation of the manuscript.

==============================
Background

Numerous studies across various populations have revealed that elevated cystatin-C levels are associated with an excessive risk of mortality. However, the prognostic value of cystatin-C remains unidentified in hypertrophic cardiomyopathy (HCM) patients. The objective of this study was to investigate whether serum cystatin-C could predict all-cause mortality independently in HCM patients.

Methods

Data from 456 HCM patients treated at West China Hospital were collected and stratified into two groups based on the median baseline serum cystatin-C level. All-cause mortality was the primary outcome. Cox regression models were used to analyze the association between cystatin-C levels and mortality risk.

Results

A total of 90 deaths were recorded over a median follow-up period of 4.67 years. Patients with higher cystatin-C levels had an increased risk of all-cause mortality (adjusted hazard ratio (HR): 2.11, 95% CI [1.30–3.42], p = 0.003) compared to those with lower levels. Time-dependent area under the curves (AUC) of cystatin-C in different time points, ranging from initial measurement to follow-up, showed a relatively stable fluctuation between 0.70 and 0.80. In comparison, the commonly used renal function markers, estimated glomerular filtration rate (eGFR) and serum creatinine, yielded lower AUC values. Restricted cubic spline curves showed that with median value of cystatin-C (1.01 mg/L) as reference, there was a gradual rise in risk of all-cause mortality with cystatin-C increasing. Subgroup analyses in female, in the patients ≥ 58 years old, and in the patients with eGFR ≥ 60 mL/min/1.73 m2 consistently confirmed robustness of the main findings.

Conclusion

Elevated serum cystatin-C levels are associated with a higher risk of all-cause mortality in HCM patients, providing valuable prognostic information beyond traditional renal function markers such as eGFR and serum creatinine.

Introduction

Hypertrophic cardiomyopathy (HCM) is the most prevalent inherited cardiovascular disorder, affecting approximately 1 in 500 individuals (Maron et al., 1995). It is characterized by marked thickening of the left ventricular wall in the absence of significant external load (Hughes & McKenna, 2005). Early reports indicated that HCM was associated with poor clinical outcomes, including heart failure, atrial and ventricular arrhythmias, sudden cardiac death (SCD), and thromboembolic events (Liu et al., 2021). However, recent advances in treatment have led to improvements in the prognosis of HCM patients, likely reflecting the development of more effective therapeutic strategies (Maron et al., 2015). Despite these advancements, a comparison of HCM mortality rates to those of the general population, based on age- and sex-stratified European mortality data, reveals that HCM mortality remains higher (Lorenzini et al., 2020). This underscores the urgent need for a more nuanced understanding of risk stratification and mortality in HCM patients.

Renal dysfunction was reported to be a common comorbidity in HCM patients (Huang et al., 2021). Chronic renal dysfunction, typically assessed by serum creatinine levels or estimated glomerular filtration rate (eGFR), is a well-established risk factor for cardiovascular disease and mortality (Chen et al., 2023; Matsushita et al., 2010). However, cystatin-C has been shown to provide superior discrimination of kidney function compared to serum creatinine, particularly in distinguishing normal from impaired renal function (Dharnidharka, Kwon & Stevens, 2002). Moreover, using cystatin-C as a biomarker for chronic kidney disease (CKD) may yield more accurate risk estimates and offer enhanced prognostic value for cardiovascular events and overall mortality, especially in low-risk populations (Rothenbacher et al., 2020).

Cystatin-C is a 13-kDa endogenous cysteine protease inhibitor produced by nearly all human cells and released into the bloodstream. Over 99% of cystatin-C is eliminated from circulation via glomerular filtration and tubular reabsorption (Mussap & Plebani, 2004). Accumulating evidence suggests that elevated cystatin-C levels are associated with several cardiovascular conditions, including coronary heart disease (Luc et al., 2006), stroke (Ni et al., 2007; Shlipak et al., 2005), myocardial infarction (Shlipak et al., 2005), heart failure (Sarnak et al., 2005), and hypertension (Kestenbaum et al., 2008). However, the clinical significance of cystatin-C as a prognostic marker in HCM patients has not been extensively studied. Therefore, this study aims to investigate whether elevated cystatin-C levels are associated with an increased risk of all-cause mortality in a cohort of hospitalized HCM patients.

Materials & Methods

Study population

This study was a retrospective, single-center, longitudinal cohort analysis registered at http://www.chictr.org.cn (registration ID: ChiCTR2000029352). Between December 2008 and March 2018, 538 adult patients diagnosed with HCM were enrolled from the cardiology inpatient department at West China Hospital of Sichuan University, Chengdu, China. Patient demographics, clinical data, laboratory results, and treatment records were comprehensively documented in the database. Following 2014 guidelines of the European Society of Cardiology (Elliott et al., 2014), the diagnosis of HCM was based on a wall thickness ≥ 15 mm (or ≥13 mm for patients with a family history of HCM) in at least one myocardial segment of the left ventricle as measured by echocardiography or cardiac MRI and where the thickness could not be explained solely by loading conditions.

At baseline, 39 patients were excluded for invalid data (including demographic information, clinical, laboratory, and treatment data, n = 34) or age below 18 years old (n = 5). In terms of loss to follow-up, another 43 patients were excluded. Finally, a total of 456 adult patients were included in the present analysis (Fig. 1).

Figure 1 Study flow diagram.

Five hundred and thirty-eight patients were diagnosed with HCM in December 2008 to March 2018. After excluding those with invalid data (n = 34) and under 18 years old (n = 5) at baseline, the remaining 499 patients were included. Due to 43 patients lost to follow-up, eventually, 456 patients with complete information were enrolled in the present analysis. HCM, hypertrophic cardiomyopathy.

The study complied with the principles of the Declaration of Helsinki and was approved by the Biomedical Research Ethics Committee, West China Hospital of Sichuan University (approval number: 2019-1147). Individual patient consent was collected. Other detailed information about those patients has been reported elsewhere (He et al., 2019; Liao et al., 2020a).

Measurement and data collection

A standard two-dimensional transthoracic echocardiography evaluation was conducted by experienced doctors in the baseline for the whole cohort during the patients’ first hospitalization from 12/2008 to 03/2018 according to the recommendations of the American Society of Echocardiography and European Association of Echocardiography (Lang et al., 2015). Venous blood specimens collected at the time of hospital admission were sent for tests (Cobas 8000 c702 module) to obtain blood biochemicals. The reference range of serum cystatin-C in electronic medical records in our hospital was 0.51−1.09 mg/L. Baseline eGFR was determined according to patients’ serum creatinine (MDRD formula and CKD-EPI formula) (Levey et al., 2006; Levey et al., 2009).

Baseline characteristics including demographic information, clinical, laboratory, and treatment data were collected from the patients’ electronic medical records in our hospital by experienced physicians. The twice-entry method was used for data entry: when the information of the two entries was consistent, the data would enter the database, otherwise it would be checked. In the present study, we picked data of basic information, medical history, family history, treatment, as well as examination including blood urea nitrogen (BUN), cystatin-C, creatinine, uric acid, triglyceride, total cholesterol, high density lipoprotein cholesterol (HDL-C), low density lipoprotein cholesterol (LDL-C), left atria, left ventricular end-diastolic dimension (LVEDD), interventricular septum, left ventricular posterior wall (LVPW), and left ventricular outflow track obstruction.

Endpoint events and follow-up

In this study, all-cause mortality was defined as the endpoint. Follow-up was conducted through clinical consultations, review of medical records, and telephone interviews, spanning from the initial evaluation to either the occurrence of the endpoint or the administrative censoring date, which was set for February 7, 2020.

Statistical analysis

Baseline serum cystatin-C was divided into two groups by the median value for the clear risk stratification. For descriptive results, variables were expressed as the mean ± standard deviation (SD), median and interquartile range, or counts (n) and percentages (%) as appropriate. Baseline characteristic differences between the two groups were analyzed using independent two-sample t-tests for variables with a normal distribution and Wilcoxon rank-sum tests for non-normally distributed variables. Interactions among categorical variables were assessed using either the chi-square test or Fisher’s exact test, depending on suitability.

Cumulative incidences were presented by Kaplan–Meier curves with log-rank tests used for comparison between groups. Univariable and multivariable Cox proportional hazard regression models were constructed to assess the relationship between serum cystatin-C and all-cause mortality. To ensure parsimony of the multivariable models, a forward stepwise approach was considered to keep significant covariates showing a univariable relationship (p < 0.100) with all-cause mortality. In multivariable analyses, models 1-5 were constructed to adjust the covariables of different categories with age and gender forced to fit the models: (1) Model 1, the basic model, adjusted for age and gender; (2) Model 2 adjusted for the basic model plus medical history including chronic obstructive pulmonary disease (COPD), prior thromboembolism events, New York Heart Association (NYHA) III/IV, atrial fibrillation; (3) Model 3 adjusted for the basic model plus medicine including warfarin, beta-blocker, and diuretic; (4) Model 4 adjusted for the basic model plus laboratory measurements including BUN, estimated glomerular filtration rate (eGFR), creatinine, uric acid, triglyceride, total cholesterol, high density lipoprotein cholesterol, and low density lipoprotein cholesterol; (5) Model 5 adjusted for the basic model plus echocardiographic data including left atria, left ventricular end-diastolic diameter (LVEDD), left ventricular posterior wall (LVPW) and ejection fraction. The final model (Model 6) included all the covariates in Model 1-5. The proportional hazards assumption with the help of the cox.zph() function shipped with the survival package or Schoenfeld’s global test in R-program was tested when a Cox proportional hazard regression model was fitted. Time-dependent multivariable adjusted Cox proportional hazards models were constructed in case the effect of the covariate was time-varying with the proportional hazards assumption of the Cox regression model not holding (Zhang et al., 2018).

Variance inflation factor (VIF) values and Pearson’s correlation coefficients were used to evaluate the degree of multicollinearity among the independent variables. A high correlation of the variables was defined as a VIF of >5 (Kim, 2019). A correlation coefficient of <0.7 between two independent variables indicates no multicollinearit (Liao, Yin & Fan, 2020b). Restricted cubic spline analysis with four knots at the 5th, 35th, 65th, 95th percentiles of serum cystatin-C was used to visualize the relationships with risk of all-cause mortality after adjustment for covariates: where there was evidence of non-linearity, a two-line piecewise linear model with a single change point was estimated by trying all possible values for the change point and choosing the value with highest likelihood. Time-dependent area under the curves (AUC) at each time of the follow-up was plotted to evaluate the accuracy of serum cystatin-C in the prediction of all-cause mortality (Kamarudin, Cox & Kolamunnage-Dona, 2017). A generally accepted approach suggests that an AUC of less than 0.60 reflects poor discrimination; 0.60 to 0.75, possibly helpful discrimination; and more than 0.75, clearly useful discrimination (Alba et al., 2017). Serum creatinine and eGFR are common kidney biomarkers in routine clinical use and have independent associations with mortality (Mooney et al., 2019; Sundin et al., 2017), so we compared the predictive ability of serum cystatin-C with that both of eGFR and serum creatinine.

Subgroup analyses with interaction tests were performed to examine whether the relationship differed by sex, age, or eGFR.

For statistical analysis, R version 3.6.3 was used. All tests were two-sided, and p values less than 0.05 were regarded as statistically significant.

Results

Baseline characteristics

The present study consists of 456 (54.61% males) adult HCM patients (Fig. 1) with an average age of 55.78 ± 15.53 years. The values of serum cystatin-C at baseline in our study varied from 0.00 to 9.49 mg/L with the median estimated at 1.01 mg/L (interquartile range: 0.89−1.17). Table 1 summarizes the baseline characteristics, comparing the two groups stratified by cystatin-C levels. Patients in the higher serum cystatin-C group were older, more likely to suffer from several comorbidities, including New York Heart Association (NYHA) III/IV, hypertension, vascular diseases, and atrial fibrillation. The uses of warfarin, statins, angiotensin-converting enzyme inhibitors (ACEIs) and angiotensin II receptor blockers (ARBs), and diuretics were also more common in the higher serum cystatin-C group. Additionally, left atrial diameter, LVPW, BUN, serum creatinine, and uric acid were found to increase significantly in the higher serum cystatin-C group, while the eGFR and ejection fraction decreased significantly.

Table 1 Baseline characteristics of studied patients between two groups.

Variables	Whole (n = 456)	Lower (n = 224)	Higher (n = 232)	p	
Basic information	
Age (year)	58.00 (46.00–67.00)	49.50 (41.00–61.00)	64.00 (53.00–72.00)	<0.001	
Gender (male)	249 (54.61%)	121 (54.02%)	128 (55.17%)	0.878	
Smoking	153 (33.55%)	80 (35.71%)	83 (35.78%)	0.989	
Drinking	76 (16.67%)	38 (16.96%)	38 (16.37%)	0.867	
Medical history	
NYHA III/IV	160 (35.09%)	63 (28.12%)	97 (41.81%)	0.003	
Hypertension	147 (32.24%)	47 (20.98%)	100 (43.10%)	<0.001	
Diabetes	38 (8.33%)	15 (6.70%)	23 (9.91%)	0.283	
COPD	27 (5.92%)	8 (3.57%)	19 (8.19%)	0.059	
Vascular disease	36 (7.89%)	9 (4.02%)	27 (11.64%)	0.004	
Prior thromboembolism event	22 (4.82%)	9 (4.02%)	13 (5.60%)	0.568	
Atrial fibrillation	81 (17.76%)	25 (11.16%)	56 (24.14%)	<0.001	
LVOTO	198 (43.42%)	97 (43.30%)	101 (43.53%)	0.960	
Family history of HCM	40 (8.77%)	21 (9.38%)	19 (8.19%)	0.778	
Family history of SCD	15 (3.29%)	8 (3.57%)	7 (3.02%)	0.945	
Medications/devices/procedures	
Aspirin/clopidogrel	101 (22.15%)	41 (18.30%)	60 (25.86%)	0.067	
Warfarin	42 (9.21%)	14 (6.25%)	28 (12.07%)	0.047	
Statin	131 (28.73%)	49 (21.88%)	82 (35.34%)	0.002	
Beta-blocker	328 (71.93%)	167 (74.55%)	161 (69.40%)	0.262	
ACEI/ARB	91 (19.96%)	28 (12.50%)	63 (27.16%)	<0.001	
Calcium channel blocker	62 (13.60%)	25 (11.16%)	37 (15.95%)	0.176	
Diuretic	95 (20.83%)	32 (14.29%)	63 (27.16%)	0.001	
Echocardiographic data	
LVEDD (mm)	43.00 (40.00–46.00)	42.00 (39.00–46.00)	43.00 (40.00–47.00)	0.136	
Left atria (mm)	40.0 (35.0–45.0)	39.0 (34.75–44.0)	41.0 (37.0–45.0)	0.002	
Interventricular septum (mm)	19.00 (16.00–22.00)	19.00 (17.00–22.00)	19.00 (16.00–22.00)	0.415	
LVPW (mm)	11.00 (10.00–13.00)	11.00 (9.00–12.00)	11.00 (10.00–13.00)	0.006	
Ejection fraction (%)	68.00 (63.00–72.00)	69.00 (65.00–73.00)	68.00 (62.00–72.00)	0.031	
Laboratory examination	
BUN (mmol/L)	6.04 (5.01–7.76)	5.70 (4.60–6.50)	6.74 (5.50–9.00)	<0.001	
eGFR (ml/min/1.73m2 )	78.05 (64.77–94.87)	89.65 (76.00–101.35)	67.45 (55.30–82.10)	<0.001	
Creatinine (umol/L)	80.00 (67.00–94.62)	73.00 (61.98–83.08)	87.95 (74.30–107.45)	<0.001	
Uric acid (umol/L)	363.00 (299.08–436.00)	338.00 (284.75–417.75)	392.50 (319.38–458.88)	<0.001	
Triglyceride (mmol/L)	1.25 (0.94–1.87)	1.28 (0.97–1.93)	1.19 (0.92–1.85)	0.268	
Total cholesterol (mmol/L)	4.29 (3.54–4.83)	4.36 (3.63–4.82)	4.22 (3.49–4.87)	0.339	
HDL-C (mmol/L)	1.27 (1.03–1.55)	1.28 (1.05–1.55)	1.25 (1.00–1.55)	0.294	
LDL-C (mmol/L)	2.42 (1.83–2.91)	2.46 (1.90–2.95)	2.36 (1.80–2.88)	0.131	
Notes.

Abbreviations ACEI angiotensin-converting-enzyme inhibitor

ARB angiotensin receptor blocker

BUN blood urea nitrogen

COPD chronic obstructive pulmonary disease

eGFR estimated glomerular filtration rate

HCM hypertrophic cardiomyopathy

HDL-C high density lipoprotein cholesterol

LDL-C low density lipoprotein cholesterol

LVEDD left ventricular end-diastolic dimension

LVPW left ventricular posterior wall

LVOTO left ventricular outflow track obstruction

NYHA New York Heart Association

SCD sudden cardiac death

Outcome

During a median follow-up period of 4.67 years (range: 0.10–10.75 years), 90 patients (19.74%) died. Of these, 64 deaths were attributed to cardiovascular causes, including heart failure (n = 26), stroke (n = 11), sudden cardiac death (SCD) (n = 25), myocardial infarction (n = 1), and hypertension (n = 1). Mortality rates per 100 person-years were 2.14 (95% CI [1.31–2.97]) and 6.75 (95% CI [5.17–8.34]) in lower serum cystatin-C group and higher serum cystatin-C group, respectively (p < 0.001).

Prognostic value of serum cystatin-C

Kaplan–Meier curves show significantly increased cumulative incidence of all-cause mortality in higher serum cystatin-C group with log-rank p < 0.001 (Fig. 2). As Table 2 shows, the crude hazard ratio (HR) for all-cause mortality in the higher serum cystatin-C group was 3.08 (95%CI [1.94–4.89], p < 0.001) compared with the lower serum cystatin-C group. Baseline variables showing a univariable relationship with all-cause mortality (p < 0.100, presented in Table S1) were chosen into multivariable models. The association that higher serum cystatin-C group had significantly increased risk of all-cause mortality remained stable and consistent in Model 1-5 (all p < 0.010). The same was true in the final model (Model 6) that higher serum cystatin-C group was associated with 2.11-fold (adjusted HR: 2.11, 95% CI [1.30–3.42], p = 0.003) risk of all-cause mortality compared with lower serum cystatin-C group after adjusting potential covariates. As Fig. 3 shows, the values of time-dependent AUC of serum cystatin-C at different time points mainly fluctuated between 0.70 and 0.80, indicating an important prognostic role in predicting all-cause mortality.

Figure 2 Cumulative incidence of all-cause mortality between serum cystatin-C groups during follow-up.

Kaplan–Meier curves show the patients in higher serum cystatin-C group had significantly higher cumulative incidence of all-cause mortality (log-rank p < 0.001) during the follow-up. HCM, hypertrophic cardiomyopathy.

Table 2 Multivariate Cox’s proportional hazard models for all-cause mortality in the whole cohort.

Model	All-cause mortality	
	Lower (n = 224)	Higher (n = 232)	
Unadjusted HR, 95% CI, p	1	3.08 (1.94–4.89), <0.001	
Adjusted HR, 95% CI, p			
Model 1	1	2.47 (1.52–4.03), <0.001	
Model 2	1	2.32 (1.44–3.75), 0.001	
Model 3	1	2.63 (1.64–4.21), <0.001	
Model 4	1	2.05 (1.24–3.38), 0.005	
Model 5	1	2.35 (1.45–3.81), 0.001	
Model 6	1	2.11 (1.30–3.42), 0.003	
Notes.

Baseline variables that showed a univariable relationship with mortality (p < 0.100) were entered for multivariable models with forward conditional stepwise regression: (1) Model 1, the basic model, adjusted for age and gender; (2) Model 2 adjusted for the basic model and medical history including COPD, prior TE, NYHA III/IV, AF; (3) Model 3 adjusted for the basic model and medicine including warfarin, beta-blockers, and diuretics; (4) Model 4 adjusted for the basic model and laboratory measurements including BUN, eGFR, creatinine, uric acid, triglyceride, total cholesterol, HDL-C, and LDL-C; (5) Model 5 adjusted for the basic model and echocardiographic data including LA, LVEDD, LVPW and EF. Model 6 included all covariables from models 1 to 5.

Abbreviations AF atrial fibrillation

BUN blood urea nitrogen

CI confidence interval

COPD chronic obstructive pulmonary disease

EF ejection fraction

eGFR estimated glomerular filtration rate

HCM hypertrophic cardiomyopathy

HDL-C high density lipoprotein cholesterol

LA left atria diameter

LDL-C low density lipoprotein cholesterol

LVEDD left ventricular end-diastolic dimension

LVPW left ventricular posterior wall

LVOTO left ventricular outflow track obstruction

NYHA New York Heart Association

SCD sudden cardiac death

TE thromboembolism event

Figure 3 Time-dependent AUCs of serum cystatin-C for predicting all-cause mortality with the extension of time.

Time-dependent AUCs of serum cystatin-C at different time points almost fluctuated between 0.70 and 0.80. AUC, area under the curves; CI, confidence interval.

A positive association between serum cystatin-C and risk of all-cause mortality could be seen after adjustment for covariates by restricted cubic spline analysis (p for non-linearity = 0.530). The median value of serum cystatin-C in the present study was 1.01, above which there was a gradual rise in risk of all-cause mortality with cystatin-C increasing (Fig. 4). While below 1.01 the association was insignificant. Based on Figs. 3 and 4, in the clinical context, baseline higher cystatin-C (for example, ≥ 1.01 mg/L) can predict a worse prognosis to a large extent at various time intervals afterward.

Figure 4 Visual associations between serum cystatin-C and risk of all-cause mortality.

Restricted cubic spline curves were used to visualize the associations between serum cystatin-C and all-cause mortality after adjustment for covariates in Model 6. Reference point was the median value at 1.01 mg/L. Dotted line represents hazard ratio at 1.

Additional analyses

In contrast, neither eGFR nor creatinine remained independently associated with all-cause mortality in the final model, let alone the time-dependent AUCs, both of which were assessed mainly below 0.70 and even 0.65 (Figs. S1–S3), showing less accurate predictive strengths than cystatin-C. Other covariates including NYHA III/IV, COPD, diuretic, BUN, total cholesterol, LVEDD, left atria, LVPW, and ejection fraction also showed independent associations with all-cause mortality. All the correlation coefficients between pairs of independent variables were <0.7 (Fig. S4) and the VIF values were close to 1, indicating no collinearity among the independent variables.

Subgroup analyses

Table 3 presents the analyses in different subgroups. Interaction tests indicated the relationship between cystatin-C group and all-cause mortality was not affected by sex, age, or GFR. Similar results to the main findings could be found, although in certain subgroups higher cystatin-C showed no significant difference from lower cystatin- C in all-cause mortality risk. Higher cystatin-C remained to be independently associated with all-cause mortality in female, in the patients ≥ 58 years old, and in the patients with eGFR ≥ 60 mL/min/1.73 m2. The values of time-dependent AUC for all-cause mortality of the three subgroups mainly fluctuated between 0.70 and 0.80, which was consistent with the overall. Moreover, an overall ascending trend of time-dependent AUC in the subgroup consisting of patients with eGFR ≥ 60 mL/min/1.73 m2 could be seen, indicating the predictive strength of serum cystatin-C was strengthening with the extension of time (Fig. S5).

Discussion

This study is the first to assess the relationship between serum cystatin-C and all-cause mortality in patients with hypertrophic cardiomyopathy (HCM). The main findings are as follows: (1) Serum cystatin-C was significantly associated with all-cause mortality, demonstrating its utility as an independent predictor of mortality in HCM patients. (2) Patients with elevated cystatin-C levels had a 2.11-fold higher risk of mortality compared to those with lower levels. (3) A positive correlation was observed between increased cystatin-C levels (above 1.01 mg/L) and increased mortality risk. (4) Time-dependent area under the curve (AUC) analysis underscored the prognostic significance of serum cystatin-C in predicting all-cause mortality. (5) Serum cystatin-C outperformed other commonly used renal function markers, including eGFR and serum creatinine, in predicting mortality risk. (6) The findings were consistent across various subgroups, including females, patients aged ≥58 years, and those with eGFR ≥60 mL/min/1.73 m2, highlighting the robustness of these results.

Renal dysfunction was common in HCM patients (Higuchi et al., 2020; Huang et al., 2021). In the present study, 18.20% of participants had an eGFR <60 mL/min/1.73 m2, which is consistent with previous reports indicating a high prevalence of kidney dysfunction in this population (Higuchi et al., 2020). Chronic kidney disease is a well-established risk factor for cardiovascular morbidity and mortality, with its detrimental effects exacerbating as renal dysfunction progresses. Furthermore, cardiovascular diseases often exacerbate renal function deterioration, creating a vicious cycle. HCM itself has also been identified as a strong predictor of the development of end-stage renal disease (Lee et al., 2019) and mortality (Huang et al., 2021). While cystatin-C is commonly used as a marker for renal function (Mussap & Plebani, 2004), our findings suggest that elevated cystatin-C levels may reflect an increased mortality risk that is partly mediated through renal dysfunction.

Table 3 Subgroup analyses by potential modifiers of association between cystatin-C and all-cause mortality.

Subgroups	Number	All-cause mortality	p for interaction	
		Lower
(Ref)	Higher
(HR, 95% CI, p)		
Sex	0.598	
Male	249	1	1.80 (0.89–3.62), 0.102		
Female	207	1	2.73 (1.34–5.56), 0.006		
Age	0.103	
<58	226	1	1.01 (0.97–1.06), 0.555		
≥ 58	230	1	3.04 (1.46–6.35), 0.003		
eGFR	0.283	
<60 mL/min/1.73 m2	83	1	0.25 (0.05–1.19) ,0.081		
≥ 60 mL/min/1.73 m2	373	1	2.10 (1.18–3.75), <0.001		
Notes.

Abbreviations CI confidence interval

eGFR estimated glomerular filtration rate

HR hazard ratio

Ref reference

However, in most clinical practice the risk of mortality is not entirely captured by measures of renal function routinely used, given that calculation of eGFR was creatinine-based. In our study, neither eGFR nor serum creatinine were independently associated with all-cause mortality. Previous studies have shown that cystatin-C is a more reliable marker for predicting mortality risk than serum creatinine (Lees et al., 2019; Shlipak et al., 2005), and our results are consistent with these findings. Both creatinine and creatinine-based eGFR demonstrated weaker predictive power for mortality risk compared to cystatin-C. These results suggest that cystatin-C may offer a more accurate interpretation of renal function and, consequently, a better prognostic assessment of mortality risk in the HCM population.

More than that, a previous study reported that cystatin-C in the highest quartile could predict all-cause mortality among participants with higher eGFR (>60 mL/min/1.73 m2) and among participants without microalbuminuria (Ix et al., 2007). It was the same case in our subgroup analysis excluding HCM patients with eGFR <60 mL/min/1.73 m2. The possible reason might be that in most clinical settings including our study, eGFR was calculated using serum creatinine, while creatinine is affected by many other factors, such as age, sex, muscle mass, diet, and physical activity, and therefore is less sensitive to mild reductions in GFR in comparison with cystatin-C (Hsu, Chertow & Curhan, 2002). Superior to serum creatinine, cystatin-C has progressively become very popular in the nephrological community for its promising role as a new, sensitive biochemical marker of changes in GFR (Mussap & Plebani, 2004).

Several specific characteristics of our cohort warrant attention. Firstly, in the whole cohort, hypertensive patients consisted of 32.2%, a larger proportion than that reported in previous HCM studies. It is well-known that hypertension is a heavy medical and health burden in China with the rising prevalence and inadequate control (Lu et al., 2017). It was reported that among 1,738,886 Chinese adults aged 35–75 years, 44.7% have hypertension (Lu et al., 2017). Secondly, a family history of SCD or HCM is a standard risk factor for death in the HCM population, however, this did not seem to make a difference in mortality in our cohort, which may be attributed to the poor medical technology in the past and the lack of systematic management in the family members. This could explain the relatively lower proportions of family history of SCD and family history of HCM in the present study.

Several limitations of this study should be acknowledged. First, the study was conducted at a tertiary referral center in Chengdu, China, which primarily serves critically ill patients. Therefore, the sample may not fully represent the general HCM population, and these findings should be interpreted with caution. Second, the retrospective, single-center design and relatively small sample size may introduce selection and information bias. Larger multicenter prospective studies are needed to validate and extend these findings. Third, we did not include biomarkers such as BNP/NT-proBNP and cardiac troponin in our analysis due to incomplete baseline data for several patients. Previous studies have shown that cystatin-C independently predicts mortality in patients with acute heart failure, even after adjusting for covariates such as BNP and heart failure risk factors (Breidthardt et al., 2017). Similarly, cystatin-C, NT-proBNP, cardiac troponin T, and NYHA III/IV were found to be independent predictors of mortality after multivariable adjustment in an unselected cohort of patients with acute heart failure (Manzano-Fernández et al., 2009). Thus, we supposed the independent prognostic role of cystatin-C even if in the presence of BNP/NT-proBNP and cardiac troponin for multivariable adjustment in HCM patients. Fourthly, in most patients, examinations like echocardiography and laboratory tests were only performed once during the first hospitalization, potential intra- and inter-observer variability might occur.

Conclusions

Our study provides novel evidence that serum cystatin-C is an independent predictor of all-cause mortality in HCM patients, surpassing traditional renal function markers such as eGFR and serum creatinine. These findings highlight the potential of cystatin-C as a valuable biomarker for improving mortality risk stratification in this population. Future research should focus on elucidating the underlying mechanisms of this association and further investigate the clinical implications of cystatin-C in the management of HCM patients.

Supplemental Information

Supplemental Information 1 Raw data

Supplemental Information 2 STROBE checklist

Supplemental Information 3 Supplemental_materials

Additional Information and Declarations

Competing Interests

Author Contributions

Human Ethics

Data Availability

The authors declare there are no competing interests.

Lu Liu conceived and designed the experiments, performed the experiments, analyzed the data, prepared figures and/or tables, authored or reviewed drafts of the article, and approved the final draft.

Yi Zheng conceived and designed the experiments, performed the experiments, analyzed the data, prepared figures and/or tables, authored or reviewed drafts of the article, and approved the final draft.

Haiyan Ruan performed the experiments, authored or reviewed drafts of the article, and approved the final draft.

Ziqiong Wang performed the experiments, authored or reviewed drafts of the article, and approved the final draft.

Xiaoping Chen performed the experiments, prepared figures and/or tables, authored or reviewed drafts of the article, and approved the final draft.

Sen He performed the experiments, prepared figures and/or tables, authored or reviewed drafts of the article, and approved the final draft.

The following information was supplied relating to ethical approvals (i.e., approving body and any reference numbers):

The study complied with the principles of the Declaration of Helsinki and was approved by the Biomedical Research Ethics Committee, West China Hospital of Sichuan University (approval number: 2019-1147).

The following information was supplied regarding data availability:

Raw data is available in the Supplemental Files.

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
