# Peer review of "Serum cystatin-C and all-cause mortality in patients with hypertrophic cardiomyopathy: a retrospective cohort study"

_PeerJ, doi:10.7717/peerj.19631_

## Round 0.1 · original submission · Major Revisions

The study entitled “Serum cystatin-C and all-cause mortality in patients with hypertrophic cardiomyopathy: a retrospective cohort study” demonstrated excellent findings using an appropriate methodological approach. However, some important points must be clarified in the manuscript. Your article has great potential for publication on PeerJ, but the reviewers have requested substantial changes to be made, mainly in methodology and discussion sessions.

Reviewer 1 ·

Basic reporting

In the manuscript titled "Serum cystatin-C and all-cause mortality in patients with hypertrophic cardiomyopathy: a retrospective cohort study," Lu Liu et al. observed that higher serum levels of cystatin-C are associated with an increased risk of all-cause mortality in patients with hypertrophic cardiomyopathy. Lu Liu et al. claimed that serum cystatin-C provides valuable prognostic information for mortality risk stratification, surpassing both eGFR and serum creatinine. This is generally an interesting story; however, some comments and suggestions are listed below:

This manuscript is readable, however, an improvement in writing would make it more scientific and understandable. See the following examples:
1. In lines 50-51, please clarify what "general population" refers to.
2. In lines 53-56, references are needed for the statement "It is well-known that cystatin-C levels correlate with various diseases and mortality."
3. In lines 59-60, "'is cleared from the circulation' is not a scientific way to describe the process."
4. In lines 242-243, I do not understand the sentence "However, in most clinical practice the risk of mortality is not entirely captured by measures of renal function routinely used in that calculation of eGFR was creatinine-based." Do you mean: "However, in most clinical practices, the risk of mortality is not entirely captured by the measures of renal function that are routinely used, given that the calculation of eGFR is based on creatinine."
5. I would recommend asking a fluent speaker outside of your research field to make sure the manuscript is clear and understandable.
6. I would also suggest reorganizing the introduction and discussion sections. Currently, the discussion is very long and contains background information that should be presented in the introduction for readers, especially those not in the cardiac field, to better understand. For example, some previous studies mentioned in lines 242-280 regarding "Renal dysfunction and HCM" can be moved to the introduction.
7. Also, I would suggest not mentioning the detailed numbers again in the discussion, such as in lines 229-241, since you have mentioned them in the result section already.

Experimental design

1. In lines 84-85, "Individual patient consent was not required due to the retrospective nature of the study." I do think patient consent or any type of proof showing that the patients are aware and agree to participate in the study is needed, even for retrospective studies.
2. The study only includes HCM patients without any control patients. It may be okay if you are comparing HCM patients with high cystatin-C to HCM patients with low cystatin-C. However, for comparisons in Table 1, including control non-HCM patients is beneficial.

Validity of the findings

1. Lu Liu et al. claim that cystatin-C provides better prognostic information for mortality risk stratification compared with eGFR and serum creatinine. However, the eGFR and serum creatinine data are not shown in the manuscript (line 197…). Including these data and making performance comparisons is important to support the claim.
2. In lines 240-241, Lu Liu et al. claimed that the "association between higher levels of cystatin-C and increased mortality in HCM patients might be partly mediated by renal dysfunction." Would it be meaningful to cluster the HCM patients based on renal dysfunction (eGFR or serum creatinine) and test the association between cystatin-C and mortality?

Additional comments

1. Supplementary Figures are in .eps format, which cannot be open unless in Adobe Photoshop.
2. No title or figure legend for supplementary figures.

·

Basic reporting

-

Experimental design

-

Validity of the findings

-

Additional comments

Chen and colleagues presented an interesting study: Serum cystatin-C and all-cause mortality in patients with hypertrophic cardiomyopathy: a retrospective cohort study.
The authors addressed an important topic, enrolled more than 450 patients with HCM, used appropriate methodology, and presented their study’s limitations and some exciting results and conclusions.
However, closer’s look has raised some comments:
1. It is not clearly presented how many Echo and MRI or both were performed on patients or on the same patient.
2. Intra- and inter-observer variability should be mentioned and discussed.
3. There is a lack of data on alcohol consumption.
4. Regarding drugs, were there no cases of MRA and SGLT2 being prescripted?
5. The CKD-EPI formula validated for Chinese population would be more accurate for eGFR.
6. There is a lack of data on malignances, the use of corticosteroids and thyroid diseases.
7. Figure 4 need a simple and relevant explanation for clinicians.

Authors should accept and discuss the comments.

---

## Round 0.2 · accepted · Accept

Dear Author,

Congratulations! After your diligent work addressing the reviewers' comments, I am pleased to inform you that your manuscript has been accepted for publication in PeerJ. This version is more concise and formal, enhancing clarity and flow.

Reviewer 1 ·

Basic reporting

All my concerns have been addressed.

Experimental design

All my concerns have been addressed.

Validity of the findings

All my concerns have been addressed.